# The implementation of a standardized optimal procedure for peripheral venous catheters' management: Results from a multi-dimensional assessment

**Fabrizio Schettini**[1☉], **Lucrezia Ferrario**[1☉]*, **Emanuela Foglia**[1☉], **Elisabetta Garagiola**[1☉], **Lionello Parodi**[2‡], **Paolo Cavagnaro**[2‡], **Luca Garra**[2‡], **Antonella Valeri**[2‡], **Monica Cirone**[2‡], **Roberta Rapetti**[2‡]

1 Centre for Health Economics, Social and Health Care Management, LIUC-Università Cattaneo, Castellanza, Italy, 2 ASL 2, Savona, Italy

☉ These authors contributed equally to this work.
‡ LP, PC, LG, AV, MC and RR also contributed equally to this work.
* lferrario@liuc.it

**Data Availability Statement:** All relevant data are within the paper and its Supporting information files.

## Abstract

Peripheral Venous Catheter (PVC) is a widely used device in the hospital setting and is often associated with significant adverse events that may impair treatment administration and patient health. The aim of the present study is to define the incremental benefits related to the implementation and the standardized and simultaneous use of three disposable devices for skin antisepsis, infusion, and cleaning, assuming the hospital's point of view, from an effectiveness, efficiency, and organizational perspective. For the achievement of the above objective, real-life data were collected by means of an observational prospective study, involving two hospitals in the Liguria Region (Northern Italy). Consecutive cases were enrolled and placed into two different scenarios: 1) use of all the three disposable devices, thus representing the scenario related to the implementation of a standardized optimal procedure (Scenario 1); 2) use of only one or two disposable devices, representing the scenario related to not being in a standardized optimal procedure (Scenario 2). For the definition of effectiveness indicators, the reason for PVC removal and the PVC-related adverse events occurrence were collected for each patient enrolled. In addition, an activity-based costing analysis grounded on a process-mapping technique was conducted to define the overall economic absorption sustained by hospitals when taking in charge patients requiring a PVC. Among the 380 patients enrolled in the study, 18% were treated with the standardized optimal procedure (Scenario 1). The two Scenarios differed in terms of number of patients for whom the PCV was removed due to the end of therapy (86.8% *versus* 39.40%, p-value = 0.000), with a consequent decrease in the adverse events occurrence rate. The economic evaluation demonstrated the sustainability and feasibility of implementing the standardized optimal procedure specifically related to the need for lower economic resources for the hospital management of adverse events occurred (€19.60 *versus* €21.71, p-value = 0.0019). An organizational advantage also emerged concerning an overall lower time to execute all the PVC-related activities (4.39 *versus* 5.72 minutes, p-value = 0.00). Results demonstrate the

**Funding:** Emanuela Foglia and Lucrezia Ferrario have presented the results of the study in two Italian Conferences and one English Conference. They received hospitality from the sponsor (fee only related to transporation costs). Otherwise, it shoudl be noted that the funders had no role in study design, data collection and analysis, decision to publish, or preparation of the manuscript. We confirm that the research cost or authors' salaries are not funded by a tobacco company. The authors are not aware of any competing interests.

**Competing interests:** The authors have declared that no competing interests exist.

feasibility in the adoption of the standardized optimal procedure for PVC management, with significant advantages not only from a clinical point of view, but also from an organizational and economic perspective, thus being able to increase the overall operational efficiency of the hospitals.

## Introduction

The Peripheral Venous Catheter (PVC) is a wisely used device in the hospital setting [1], representing an essential element of modern medicine, used for the administration of fluids, nutrients, drugs, and blood products, as well as in the collection of blood for examinations [2–4]. Up to 70% of patients require a PVC during their hospital stay, and conservative estimates suggest that PVC days account for 15%-20% of total patient days in acute care hospitals [5].

Although such devices are frequently used and often considered to present a low risk for the patient, PVCs are associated with significant adverse events that may impair treatment administration and patient health [6–8]. In optimal conditions, the mean dwell time of PVCs is 3–4 days, with a median dwell time of 2 days [9–11]. The short dwell time, which approaches the recommendation of the US Centers for Disease Control and Prevention (CDC) for catheter replacement, is usually the result of short operating times and short hospital stay [10, 12]. In contrast, PVCs often fail before the therapy is completed: this can occur because the device is not adequately attached to the skin, allowing the PVC to fall out, thus leading to complications such as phlebitis, infiltration, or occlusion, with the consequent increase in patient morbidity and length of stay [13]. The development of any of the above complications would lead to the PVC's removal before the end of its intended dwell time [14, 15].

According to the above, recent studies show that, although scheduled cannula needle removal is common practice in many hospitals, regular catheter replacement does not reduce the occurrence of complications such as phlebitis, thrombophlebitis, infiltration, extravasation, occlusion, venous spasm, and catheter-related infections [15, 16]. Furthermore, it has been shown that the insertion and maintenance of PVCs conducted by untrained or inexperienced healthcare workers increases the risk of thrombophlebitis [17]. Infection and phlebitis are of primary concern [11], so registered nurses must ensure that their knowledge and skills related to the management of PVCs are up to date and evidence-based [18, 19] to reduce complications associated with these devices.

In this regard, both the choice of the appropriate medical device and appropriate knowledge of its use throughout the entire process of PVC management -from skin antisepsis to washing up -could represent independent factors for successful patient care.

According to the above, literature on the topic suggested the simultaneous use of three specific disposable medical devices, such as ChloraPrep (disposable device used for skin antisepsis), Nexiva (PVC, to be inserted), and Posiflush (disposable device to be used for staying "in situ" and for washing up the patients after the catheter removal). On the one hand, the use of ChloraPrep is implemented for skin preparation to rapidly kill micro-organisms; it also continues the antimicrobial activity for at least 48 hours [20, 21], thereby leading to a decrease in PVC-related infections [22–24]. ChloraPrep is composed of 2% chlorhexidine gluconate and 70% isopropyl alcohol in a single-dose applicator and sterile solution. On the other hand, the PVC Nexiva, in comparison to traditional cannula, incorporates a stabilization platform, an extension set and a needle-less access site, which could significantly reduce the risks of developing complications, such as phlebitis, infiltration/extravasation and dislodgement, as well as

bloodstream infections [25, 26]. Besides an increased safety profile for patients, the use of Nexiva would simultaneously offer protection to healthcare professionals, reducing the risk of exposure associated with needlestick injury [27]. Furthermore, the use of Posiflush as a medical device for washing-up activities would reduce catheter-related bloodstream infections, thus improving the standards of practice for catheter maintenance and management [28]. Posiflush is a specific pre-filled syringe containing 0.9% sodium chloride, whose main aim is to eliminate any blood reflux and maintain the patency of the cannula, while also reducing the risk of contamination. In order to guarantee the implementation of the clinical procedure as requested by the most diffuse Italian and European Guidelines [10–12, 15], the use of all three of the above-mentioned disposable medical devices (ChloraPrep, Nexiva and Posiflush) should be integrated with the following targets activities: i) number of insertion attempts lower than four; ii) PVC replacement in 96 hours; and iii) number of washes higher or equal to the number of PVC insertion. Based on the above, the use of all three disposable medical devices and the conduction of all three target activities would represent the standardized optimal procedure for PVC management.

Based on these considerations, the present study aims at analyzing the management of the PVC process in clinical practice in term of the outcomes measures achieved (from the hospital perspective) in the implementation of a standardized optimal procedure for PVC implant and management (composed of skin antisepsis, insertion and washing up activities). In addition, both the organizational and economic incremental benefits were defined to understand the potential optimization area for the hospitals taking in charge patients requiring a PVC but not having yet implemented a procedure optimizing skin antisepsis, insertion, and washing-up activities.

## Methods

For the achievement of the above objective, a Health Technology Assessment analysis was conducted by means of the Danish Mini-HTA hospital-based model [29], which is useful to explore the clinical, economic, and organizational implications related to the standardized optimal procedure for PVC management, in comparison with the current situation, where this procedure is not utilized for all patients.

Real-word data were collected during a prospective observational study. It should be noted here that the present study does not have an interventional nature: even though the standardized optimal procedure was well-known in the hospitals involved, thus being integrated in the standard clinical practice, nurses would voluntarily choose the medical devices to be used in the skin antisepsis, insertion and washing-up phases of the PVC management process. As a result, they often only partially followed the entire process or diverged completely from the standardized optimal procedure (the use of the three above-cited disposable medical devices—ChloraPrep, Nexiva and Posiflush—in the three-step phase: i) number of insertion attempts lower than four; ii) PVC replacement in 96 hours; and iii) number of washes higher or equal to the number of PVC insertion-).

This study was approved by the Ethical Committee of the hospitals involved (ASL 2 Azienda Sociosanitaria Ligure 2, Savona, Italy). The informed consent form was collected, and patients who did not sign the form were excluded from the study.

Consecutive cases of patients were enrolled within 5 Operative Units of Medical or Surgical Departments, involving two Hospitals in the Liguria Region (Italy), from September 2018 to January 2019. This occurred after approval of the Ethics Committee (ASL 2 Azienda Sociosanitaria Ligure 2, Savona, Italy) and according to the following inclusion criteria: i) age older than 18 years; ii) length of hospitalization ranging from 4 to 15 days; and iii) use of PVC.

Adult patients' consecutive cases were enrolled and placed into either a "being in a standardized optimal procedure" (Scenario 1) or a "not being in a standardized optimal procedure" group (Scenario 2), for whom demographic (age, gender) clinical (body max index-BMI, presence of comorbidities, vein status) and economic data were gathered. It should be noted here that Scenario 1 (i.e., being in a standardized optimal procedure) refers to the simultaneous use of the three disposable medical devices (ChloraPrep, Nexiva and Posiflush), integrated with the proper conduction of the previous targets' activities described and required by guidelines: (i) number of insertion attempts lower than four; ii) PVC replacement in 96 hours; and iii) number of washes higher or equal to the number of PVC insertion), for the proper conduction of skin antisepsis, insertion and washing-up phases. On the contrary, Scenario 2 (i.e., not being in a standardized optimal procedure) refers to the use of only one or two out of the three disposable medical devices (thus using different devices for antisepsis, insertion and washing-up phases), with target activities missing.

From an economic perspective, the evaluation of PVC management costs was accordingly developed by means of an Activity-Based Costing (ABC) analysis [30]. In particular, the following drivers of hospital management costs were valorized, considering the entire PVC management process and prospectively collected for each patient enrolled in the study:

i) Involvement of human resources, in terms of time for executing the PCV-related activities, valorized in accordance with the Italian National Labour Contracts per professional class; ii) number and typology of medical devices used, considering both the medical devices composing the standardized optimal procedure and other comparable medical devices; iii) consumables; iv) general and fixed costs, considering not only cleaning services and meals, but also energy, maintenance services or third party and service contracts. All the above items of healthcare expenditure were derived from accounting flows by cost center provided by the management control of the hospitals involved and were evaluated considering the purchasing costs plus related VAT.

The economic analysis assumed the hospital's perspective and estimated the hospital costs sustained up to 12 months, considering all the PVC implants performed on an annual basis.

The economic evaluation of the process was integrated with a budget impact analysis to define the economic sustainability of the overall adoption of the standardized optimal procedure [31]. To design the budget impact analysis, a baseline scenario (or base-case scenario) consisting of the real-life implementation of the PVC procedure was compared to different innovative scenarios, diverging from a different use of implementation of standardized optimal PVC procedure. Specifically, the baseline scenario considered the adoption of the standardized optimal PVC procedure for only 18% of the PVC implants, as observed considering real-world evidence from the hospitals involved in the study, evaluating an overall number of 156,624 PVC devices implanted on an annual basis. The base-case scenario was compared with three innovative scenarios where the standardized optimal PVC procedure would be implemented to an incremental portion of patients (35%, 50% and 100%).

Once having collected the above information, data were first analyzed considering descriptive statistics, frequencies, and distributions to give a comprehensive picture of the sample of reference.

After having verified the normal distribution of all the variables under assessment, independent sample T-tests were used to describe the existence of statistically significant differences between "being in a standardized optimal procedure" and "not being in a standardized optimal procedure" groups, from a demographic, clinical and economic point of view.

Finally, a hierarchical sequential linear regression model (with enter methodology), was implemented to define patterns and determinants of effectiveness and costs, using the Adjusted $R^2$ to check the explanatory power of each model [32]. According to the above,

effectiveness and costs acted as dependent variables of the model, which could be influenced by the following independent variables: i) the implementation or not of the standardized optimal PVC procedure; ii) typology of medical devices used for skin preparation, implant and washing-up activities, considering not only medical devices composing the standardized optimal procedure, but also other comparable medical devices; iii) patient's BMI; iv) vein characteristics, assessed by means of the A-DIVA Scale [33]; v) number of PVCs days in situ; vi) PCV-related activities execution time.

All statistical analyses were performed using the statistical software SPSS 22.0.

## Results

### Description of the sample

The sample under assessment was composed of 380 patients, referring to five Operating Units of two Italian Hospitals.

Out of them, only 68 patients (18%) were treated according to the above-mentioned PVC standardized optimal procedure. In the comparison among groups (being in a standardized optimal procedure–Scenario 1—*vs* not being in a standardized optimal procedure–Scenario 2), the populations under assessment are well-matched and superimposable concerning demographic and clinical indicators, since no statistically significant differences emerged (p-value > 0.05), demonstrating the possibility to compare groups both for effectiveness and cost results (Table 1).

### Results from effectiveness and safety indicators

The effectiveness measure was related to the percentage of patients for whom PVC removal was due to the end of therapy. Real-life data revealed that for patients being in a standardized optimal procedure 86.8% of the PVC removal was due to the end of the therapy and not associated to adverse events, as in Scenario 2 (39.4%, p-value = 0.000)–Table 2.

Focusing on the patients for whom the PVC removal was due to the development of adverse events (Table 3), it emerged that the implementation of the standardized optimal procedure guaranteed the lower level of occurrence for complications, considering, in particular, conditions of occlusion and phlebitis (p-value = 0.000).

While accidental displacement is an adverse event related to the implant procedure (with an incidence rate increase in case of no standardized optimal procedure implementation, p-

**Table 1. Description of the sample under assessment.**

|  | Entire Sample N = 380 | Being in a standardized optimal procedure N = 68 | Not being in a standardized optimal procedure N = 312 | p-value |
|---|---|---|---|---|
| Age [Average Value ± standard error] | 71.18 ± 0.78 | 72.48±2.03 | 70.87±0.95 | 0.476 |
| Gender—Female [%] | 55% | 52.9% | 55.4% | 0.403 |
| Body Max Index—BMI [Average Value Standard Error] | 24.99±0.19 | 24.66±0.39 | 25.06±0.22 | 0.422 |
| Presence of Comorbidities [%] | 72.90% | 72.1% | 73.1% | 0.486 |
| Vein Characteristics—visible and palpable [%] | 43.94% | 45.6% | 43.6% | 0.061 |
| Length of Hospitalization [Average Value ± standard error] | 11.69±0.37 | 11.93±1 | 11.51±0.40 | 0.674 |
| Number of attempts at PVC cannulation [Average Value ± standard error] | 2.21±0.23 | 1.92±0.11 | 2.22±0.31 | 0.314 |
| Days PVC stay in situ [Average Value ± standard error] | 7.02±0.45 | 7.91±0.61 | 6.85±0.31 | 0.098 |

**Table 2. Effectiveness indicators.**

| | PVC removal due to the end of therapy [%] | PVC removal due to the adverse events [%] |
|---|---|---|
| Being in a standardized optimal procedure–Scenario 1 | 86.8% | 13.2% |
| Not being in a standardized optimal procedure–Scenario 2 | 39.40% | 60.60% |
| p-value | 0.000 | |

**Table 3. A focus on the development of adverse events.**

| | PVC extravasation | Accidental Displacement | PVC removal by patient | Occlusion | Phlebitis |
|---|---|---|---|---|---|
| Being in a standardized optimal procedure–Scenario 1 | 0.00% | 0.00% | 8.82% | 2.94% | 1.47% |
| Not being in a standardized optimal procedure–Scenario 2 | 2.95% | 10.86% | 8.33% | 12.48% | 14.43% |
| p-value | 0.000 | 0.000 | 0.314 | 0.000 | 0.000 |

**Table 4. Regression models for effectiveness.**

| | Model 1 | Model 2 | Model 3 | Model 4 | Model 5 |
|---|---|---|---|---|---|
| Being in a standardized optimal procedure | 0.361* | 0.052 | -0.140 | -0.226* | -0.211* |
| Medical devices used for skin antisepsis | | -0.402* | -0.500* | -0.549* | -0.539* |
| PVC Typology | | | 0.327* | 0.320* | 0.312* |
| Medical devices used for washing-up activities | | | | 0.233* | 0.231* |
| BMI | | | | | -0.110* |
| $R^2$ | 0.130 | 0.197 | 0.286 | 0.336 | 0.348 |
| Adj $R^2$ | 0.128 | 0.193 | 0.280 | 0.329 | **0.340** |
| F Value | 56.541* | 46.238* | 50.156* | 47.526* | 39.984* |
| $\Delta R^2$ | **0.130** | 0.067 | **0.089** | 0.051 | 0.012 |
| F ($\Delta R$) | 56.541* | 31.389* | 46.767* | 28.592* | 6.853* |

value = 0.000), PVC removal by the patient only depends on the specific patient's clinical conditions, such as potential neurological or movement disorders. The use of the procedure would allow for a significant improvement, especially regarding a reduced occurrence of complications and adverse events, greater stability of venous access, better prevention of the risk of displacement and reduction in the number of needlestick injuries.

The above-mentioned considerations were confirmed by the multivariate analysis. The regression model for effectiveness (Table 4) revealed that the different use of technologies (specifically the use of certain types of PVCs, as well as the Posiflush), and a standard value of BMI, could predict a variability of the effectiveness of the therapy (Adj $R^2$ = 0.340), determining the achievement of a greater clinical outcome.

## Results from the economic and the organizational assessments

The economic evaluation (Table 5) shows the feasibility of the implementation of the standardized optimal PVC procedure: despite higher costs in the technology used, "being in a standardized optimal procedure" group was related to a lower overall process cost (p-value = 0.019), given the occurrence of fewer adverse events requiring both a repositioning of the PVC and the clinical management of the patient for their resolution (p-value = 0.000).

A longer duration of PVC stay in situ, a higher execution time, as well as the removal of the PVC before the end of therapy due to the occurrence of an adverse event and/or a

**Table 5. Economic evaluation of the process.**

| | Human Resources [€][a] | Cost of Accessories [€] | Cost of technology at first positioning [€][b] | Cost of technology after first positioning [€][b] | Sub-total Procedure [€] | Repositioning for adverse events [€][c] | Assessment of adverse events [€][d] | Sub-total adverse events [€] | Total Cost [€] |
|---|---|---|---|---|---|---|---|---|---|
| **Being in a standardized optimal procedure–Scenario 1** | € 4.60 | € 0 | € 5.04 | € 8.24 | **€ 17.88** | € 1.39 | € 1.05 | € 2.44 | **€ 19.60** |
| **Not being in a standardized optimal procedure–Scenario 2** | € 5.28 | € 0.79 | € 5.04 | € 7.73 | **€ 18.82** | € 3.46 | € 1.48 | € 4.93 | **€ 21.71** |
| **P-value** | 0.004 | 0.000 | 0.412 | | **0.021** | 0.000 | 0.000 | 0.000 | **0.019** |

[a] Economic evaluation of the time spent by healthcare professional along the entire PVC process, from skin antisepsis to washing up

[b] Economic evaluation of the devices used for the entire PVC process, considering all the attempts conducted for PVC insertion

[c] Economic evaluation of the devices used if an adverse event had occurred

[d] Economic evaluation of the management of adverse events, in terms of time spent by healthcare professionals as well as potential exams or procedure conducted for the hospital resolution of the complications occurred, according to the incidence rates presented in Table 3

complication, significantly determined a higher peripheral venous access process cost (Adj $R^2$ = 0.630), as detailed in Table 6.

According to a 12-month time horizon and assuming the hospital perspective (Table 7), in the comparison between Scenario A and Scenario D (moving from 18% of the standard procedure use, to 35%), hospitals could benefit from economic savings equal to 3.51% for the implantation of 156,624 PVCs on an annual basis. The more the standardized optimal procedure based on disposable devices implant is implemented, the more the economic savings become higher, ranging from 5.09% (in case of 50% of market share for innovative implant procedure–comparison between Scenario A and Scenario C) to 9.71% (in case of 100% of market share for innovative implant procedure–comparison between Scenario A and Scenario B).

The above advantage would not be relegated only to the economic sphere. The implementation of the standardized optimal procedure, given a lower execution time along the entire PVC management process (4.39 minutes *versus* 5.72 minutes, *p-value* = 0,001), would also generate significant organizational advantages, from a hospital capacity perspective.

**Table 6. Regression model for costs.**

| | Model 1 | Model 2 | Model 3 | Model 4 | Model 5 |
|---|---|---|---|---|---|
| **Vein status** | 0.158* | 0.137* | 0.066 | 0.059 | 0.046 |
| **PVC Stay in situ** | | 0.683* | 0.669* | 0.676* | 0.659* |
| **Average execution time** | | | 0.326* | 0.318* | 0.311* |
| **Being in a standardized optimal procedure** | | | | 0.062* | 0.013* |
| **PVC effectiveness** | | | | | 0.212* |
| **$R^2$** | 0.025 | 0.491 | 0.592 | 0.596 | 0.634 |
| **Adj $R^2$** | 0.023 | 0.489 | 0.589 | 0.592 | **0.630** |
| **F Value** | 9.737* | 182.001* | 182.030* | 138.250* | 129.791* |
| **$AR^2$** | 0.025 | **0.466** | **0.101** | 0.004 | 0.038 |
| **F (AR)** | 9.737* | 345.393* | 93.132* | 3.410* | 39.371* |

**Table 7. Budget impact analysis.**

| Scenarios under assessment | Total healthcare costs related to the implantation of 156,624 PVCs on annual basis |
|---|---|
| Baseline Scenario A–% of hospital standardized optimal procedure penetration equal to 18%—real life Scenario | € 1,770,719 |
| Innovative Scenario B—% of hospital standardized optimal procedure penetration equal to 100%—best case Scenario | € 1,598,871 |
| Innovative Scenario C—% of hospital standardized optimal procedure penetration equal to 50% | € 1,684,933 |
| Innovative Scenario D—% of hospital standardized optimal procedure penetration equal to 35% | € 1,710,751 |
| Δ € B-A | - € 171,848 |
| **Δ % B-A** | **- 9.71%** |
| Δ € C-A | - € 85,786 |
| **Δ % C-A** | **- 5.09%** |
| Δ € D-A | - € 59,968 |
| **Δ % D-A** | **- 3.51%** |

In this regard, considering the 156,624 PVC devices implanted on an annual basis and assuming the same scenarios of the BIA (Table 8), the organizational benefits would range from a minimum of 5.53% of time savings (in case of 35% of standardized optimal procedure introduction), to a maximum of 20.78% (in case of 100% of standardized optimal procedure implementation), in terms of overall minutes spent for all the PVC-related activities.

## Discussion

PVCs are now an essential part of medical care, and their management has an important effect on the incidence of catheter associated infections. Thus, any strategies that are able to prevent the occurrence of PVC-related infection and maximize the clinical outcomes for patients need to be deeply considered, as most complications associated with the use of PVCs are preventable [34].

**Table 8. Organizational benefits.**

| Scenarios under assessment | Total organizational advantages related to the implantation of 156,624 PVCs on annual basis |
|---|---|
| Baseline Scenario A–% of hospital standardized optimal procedure penetration equal to 18%—real life Scenario | 452,046.09 minutes |
| Innovative Scenario B—% of hospital standardized optimal procedure penetration equal to 100%—best case Scenario | 358,114.54 minutes |
| Innovative Scenario C—% of hospital standardized optimal procedure penetration equal to 50% | 412,158.02 minutes |
| Innovative Scenario D—% of hospital standardized optimal procedure penetration equal to 35% | 428,371.07 minutes |
| Δ € B-A | -93,931.54 minutes |
| **Δ % B-A** | **-20.78%** |
| Δ € C-A | -39,888.06 minutes |
| **Δ % C-A** | **-9.68%** |
| Δ € D-A | -23,675.02 minutes |
| **Δ % D-A** | **-5.53%** |

Some of the traditional preventive measures are represented by training and education of healthcare professionals and patients, performance feedback, specialized intravenous treatment teams, documentation with peripheral cannula care plans, hand hygiene, skin preparation, use of sterile semipermeable dressings, selection of catheter insertion site and catheter replacement strategies [5, 35, 36].

The present analysis could be considered the first Italian attempt to investigate the impact of a specific standardized optimal procedure for the management of PVCs, designed according to the most recent guidelines on the topic [10–12, 15], not only from an effectiveness perspective, but also considering the potential economic and organizational impacts for the hospitals taking in charge patients requiring a PVC. The clinical course of each patient who received a PVC for any reason in two Italian hospitals were closely followed by trained nurses until removal of the catheter.

The implementation of the standardized optimal procedure reported the preferable solution of a lower incidence of adverse events and a reduced number of attempts at PVC cannulation, with a consequent minimization of costs.

From an organizational point of view, a higher PVC stay in situ, and a lower execution time for the PVC procedure meant an increase in operational efficiency related to the procedure and a decrease in the nursing activities devoted to the weaker patients. From an economic perspective, the BIA reported marginal investments related to the acquisition of new technologies, strictly dependent on the baseline scenario of the hospital choosing to implement new technologies.

The above considerations are strengthened by inferential analysis conducted for the definition of the predictors of costs optimization and effectiveness maximization. On the one hand, the use of certain types of PVCs, as well as the Posiflush, and a standard value of BMI, could determine the achievement of a greater clinical outcome. On the other hand, a longer duration of PVC stay in situ, a higher execution time, as well as the removal of the PVC before the end of therapy due to the occurrence of an adverse event and/or a complication, significantly determined a higher economic resource absorption devoted to patients requiring a PVC.

Even though the main strength of the study was that it relied on real-life data, it has some limitations. The study was implemented in two hospitals and was supported by nursing leadership. In this view, a further validation in other hospitals may be needed to support the generalizability of the results, in particular concerning the different base-case use of the standardized optimal procedure, as well as the definition of the different devices used along the entire PVC management process.

In conclusion, the results of the study suggested the strategic relevance of the standardized optimal procedure for the management of PVC implementation in the improvement of the clinical pathway for patients, with important economic and organizational savings for hospitals.

## Supporting information

**S1 Data.**
(XLSX)

## Acknowledgments

The Authors of the present paper would like to sincerely thank all the healthcare professionals involved in the data collection who helped in the achievement of the study's objectives. Furthermore, the Authors would like to thank Prof. Fulvio Pinelli (Careggi University Hospital, Florence, Italy) and Prof. Mauro Pittiruti (Catholic University Hospital, Rome, Italy), who offered their vital criticism to revise the contents of this paper.

## Author Contributions

**Conceptualization:** Emanuela Foglia, Monica Cirone.

**Data curation:** Lucrezia Ferrario, Elisabetta Garagiola, Monica Cirone, Roberta Rapetti.

**Formal analysis:** Fabrizio Schettini, Elisabetta Garagiola.

**Methodology:** Emanuela Foglia.

**Project administration:** Emanuela Foglia.

**Supervision:** Luca Garra, Antonella Valeri.

**Validation:** Lionello Parodi, Paolo Cavagnaro, Luca Garra.

**Visualization:** Lionello Parodi, Paolo Cavagnaro.

**Writing – original draft:** Fabrizio Schettini, Lucrezia Ferrario.

**Writing – review & editing:** Emanuela Foglia, Roberta Rapetti.

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
