## [Decision Letter · Decision Letter 0]

31 May 2021

PONE-D-21-00993

PERIPHERAL VENOUS ACCESS PROCESS and BEYOND: RESULTS FROM THE FIELD

PLOS ONE

Dear Dr. Ferrario,

Thank you for submitting your manuscript to PLOS ONE. After careful consideration, we feel that it has merit but does not fully meet PLOS ONE’s publication criteria as it currently stands. Therefore, we invite you to submit a revised version of the manuscript that addresses the points raised during the review process.

Please address the issues and revise accordingly.

We look forward to receiving your revised manuscript.

Kind regards,

Academic Editor

PLOS ONE

Journal Requirements:

2. Please modify the title to ensure that it is meeting PLOS’ guidelines (https://journals.plos.org/plosone/s/submission-guidelines#loc-title). In particular, the title should be "specific, descriptive, concise, and comprehensible to readers outside the field" and in this case it is not informative and specific about your study's scope and methodology.

3. Thank you for including your ethics statement: "The study, whose results are here presented, was approved by the Ethical Committee of the hospitals involved (ASL 2, Savona, Italy). The informed consent form was collected, and patients who did not signed the form were excluded from the study.".   

4. We noted in your submission details that a portion of your manuscript may have been presented or published elsewhere. [Emanuela Foglia and Lucrezia Ferrario have presented the results of the study in two Italian Conferences and one English Conference. ] Please clarify whether this conference proceeding was peer-reviewed and formally published. If this work was previously peer-reviewed and published, in the cover letter please provide the reason that this work does not constitute dual publication and should be included in the current manuscript.

5. Thank you for stating in your financial disclosure: 

"Emanuela Foglia and Lucrezia Ferrario have presented the results of the study in two Italian Conferences and one English Conference.

They received hospitality from the sponsor (fee only related to transporation costs).

Otherwise, it shoudl be noted that the funders had no role in study design, data collection and analysis, decision to publish, or preparation of the manuscript."

PLOS ONE requires you to include in your manuscript further information about the funder so that any relevant competing interests can be assessed. Please respond to the following questions:

Please state whether any of the research costs or authors' salaries were funded, in whole or in part, by a tobacco company (our policy on tobacco funding is at http://journals.plos.org/plosone/s/disclosure-of-funding-sources) Please state whether the donor has any competing interests in relation to this work (see http://journals.plos.org/plosone/s/competing-interests) .Please state whether the identity of the donor might be considered relevant to editors or reviewers’ assessment of the validity of the work.If the donors have no perceived or actual competing interests, please state: “The authors are not aware of any competing interests”.

This information should be included in your cover letter. We will amend your financial disclosure and competing interests on your behalf.

7. Thank you for submitting the above manuscript to PLOS ONE. During our internal evaluation of the manuscript, we found significant text overlap between your submission and the following previously published works, some of which you are an author.

https://www.valueinhealthjournal.com/article/S1098-3015(20)33320-9/fulltext?_returnURL=https%3A%2F%2Flinkinghub.elsevier.com%2Fretrieve%2Fpii%2FS1098301520333209%3Fshowall%3Dtrue#articleInformation

Please revise the manuscript to rephrase the duplicated text, cite your sources, and provide details as to how the current manuscript advances on previous work. Please note that further consideration is dependent on the submission of a manuscript that addresses these concerns about the overlap in text with published work.

Reviewers' comments:

Reviewer's Responses to Questions

**Comments to the Author**

1. Is the manuscript technically sound, and do the data support the conclusions?

Reviewer #1: No

Reviewer #2: Partly

2. Has the statistical analysis been performed appropriately and rigorously? 

Reviewer #1: I Don't Know

Reviewer #2: Yes

3. Have the authors made all data underlying the findings in their manuscript fully available?

Reviewer #1: No

Reviewer #2: Yes

4. Is the manuscript presented in an intelligible fashion and written in standard English?

Reviewer #1: Yes

Reviewer #2: Yes

5. Review Comments to the Author

Reviewer #1: Although the article treats a relevant topic, I think it needs additional work and a careful revision to be considered for publication. Many aspects are not clear and should be improved.

- Throughout the article you mention PVCs standard procedure. Honestly, it’s not so clear what you mean by standard. Who defined this standard? Are there any clinical guidelines or hospital guidelines? Or is it just a hypothesis you want to test? In case it’s a gold standard (the word standard implicitly refers to a gold standard) you should add a reference. If it is a standard, why is it applied only to 18% of patients? Who decides whether the standard has to be applied or not?

- You define as intervention the “implementation of an innovative PVCs standard procedure” and as comparator the “standard procedure is not utilized for all the patients”. What do you mean by innovative: using the innovative BD Nexive PVC or using the so-called standard procedure (i.e. the three BD devices) for all patients? Also, in line 123 what do you mean by innovative scenario? It seems to me you are simply simulating different penetration rate of PVC.

- It’s not very clear what you mean by “being in procedure” and “not being in procedure” groups. Please explain more clearly. Also explain how the patients are assigned to the two groups: randomly? based on clinical considerations?

- When you explain the goal of the study in the introduction, you say “based on these considerations”, that seems to be linked to the choice of the appropriate medical device and presenting appropriate knowledge and skills (previous sentence). None of these elements is mentioned in the article. But then, in the regression analysis you mention PVC typology, so one can argue that nurses have to possibility to choose the type of device and the three BD devices are not used for all the patients. Please explain.

- In the abstract you mention ABC was performed, but this is not mentioned in the body of the article. The method used to collect the data in table 5 should be explained. You should also explain the meaning of each variable in table 5.

- In table 1 you mention Vein Characteristics - visible and palpable: how is this variable measured? Is there an objective measure or it relies on the nurse perception? Does the operator experience influence the evaluation of this aspect?

- Please clarify which patients’ comorbidities are considered in the analysis.

- Five wards were belonging to two different hospitals participated in the study. Did you find any differences (e.g. % of in procedure patients) among hospitals and medical vs surgical wards?

- In table 4 please make clear the dependent variable. Moreover, in the text you should explain more clearly the meaning of the coefficients: for instance, how to interpret the meaning of the coefficient of disinfection?

- What do you mean by “simulated the healthcare expenditure sustained up to 12 months?” It seems to me you mean costs born by the hospitals instead of healthcare expenditure. Purchasing cost?

- The costs that have been considered in the economic analysis are not clear. Where can I find the cost of the devices (three in the treatment group, 1 or 2 in the control)? Any costs for the management of adverse events?

- Budget impact analysis: it’s not clear to me how the numbers in table 7 were computed.

- In the economic?

- Last, I can guess that the BD products mentioned in the paper are purchased through regional tenders. In case another economic operator will award next tender, will the results be valid anyhow? On other words, are the results product-specific o procedure-specific?

Minor aspects:

- Lines 74-79: the paragraph is too long and as a result not clear.

- Not very clear what BD Chloraprep and Posiflush are.

- What do you mean by “BD Nexiva innovative PVC”? The presence of a stabilization platform? What is this?

- What do you mean by “reducing the risk of exposure associated with needlestick injury”?

- Line 102 perspective: do you mean prospective?

- Line 106: sign instead of signed (same in the abstract)

- Line 120-121 What do you mean by “assuming 156,624 PVCs”? what type of assumption is this? Is it the volume of PVCs in the 5 analyzed departments?

- You say “Real-life data revealed that, for patients being in procedure, 86.8% of the PVC removal was due to the end of the therapy, and not associated to adverse events, as in the other scenario (13.2%, p-value=0.000)” It’s not clear to me what you mean by “as in the other scenario”: 13from the table it seems that 13.2% relates to the same scenario, to removal due to adverse events.

- It’s not clear who is the sponsor of the study.

Reviewer #2: The authors used a single group of continuous measurement and observation to compare the correlation between the applied of ultrasonographic and PIVC failure or survival. This is an interesting topic, however, there are still great doubts about the rigor of the research design which influencing the internal and external validity of the study:

1.There is no control group and sample size are too small without any theoretical sample size estimation. This may also be the reason why most of the statistical results are not significantly different.

2.How consistent is the assessment of EMR and ultrasound operation between different researchers? Performing a test of internal and external consistency of the researcher could be considered.

3.Author should consider and analysis that the working years and professional abilities of ER nurses may also be the key factors that affect the placement of PIVC and the judgment of PIVC failure or survival.

4.The definition of subcutaneous edema in the image needs more precisely descript rather just presents the photo.

Some parts in the article are not clearly described:

1.There are inconsistent descriptions of PIVC failures in the text, such as line 164 and line 221, which may cause confusion for readers.

2.The primary outcome is not clear enough in line 219. What is the definition of “increased risk of PIVC failure” and how to measure it?

3.“Day idle”, a significant variable between groups in the outcome measurement, but there is no any discussion about that.

4.If the ultrasonographic shows subcutaneous edema, but the clinical standard assessment process by ER nurse is not abnormal, is it necessary to remove the PIVC preventively? Are there any other suggestions for the above condition? I don’t see any discussion for that.

In the end, I agree with the reviewer#5 point out the problem of result application. I think the suggestion is also not specific for clinical practice in ER.

6. PLOS authors have the option to publish the peer review history of their article (what does this mean?). If published, this will include your full peer review and any attached files.

Reviewer #1: No

Reviewer #2: No

---

## [Author Response · Author response to Decision Letter 0]

18 Sep 2021

Replies to the Editor

Dear Editor,

Thank you very much for your comments.

We confirm that in the amended manuscript, you could find the Ethics statements, with regard to the name of the ethics committee/institutional review board(s) that approved your specific study. In particular, the study, whose results are here presented, was approved by the Regional Ethical Committee of the hospitals involved (ASL 2 Azienda Sociosanitaria Ligure 2, Savona, Italy), according to the Liguria Region protocol number 231/2018 dated 18/10/2018. The informed consent form was collected, and patients who did not signed the form were excluded from the study.

Furthermore, we have made the following changes as requested.

- Modification of the title, in “Organizational, efficiency and effectiveness impacts of a standardized optimal procedure, for the PVCs management”.

- Inclusion of all the tables as part of the main manuscript, with the consequent removal of the individual files.

- Upload of the minimal data set used for data analysis, as a Supporting Information file.

- Integration in the cover letter of specific statement, with regard to the previous divulgation of the results presented in this manuscript.

With regard to other submissions related to the present manuscript, we confirm that it has been presented only in Italian and English informative conferences, without the requirement to publish any abstracts or full text. In particular, the audience of the above conferences were nurses and clinicians currently using the medical devices investigated, thus taking in charge patients needing a peripheral venous access. According to the above, we confirm that no conference proceedings emerged, and no documents related to the present topic was reviewed and formally published.

Besides the above conferences, we have presented an abstract related to the results of the present research activity to ISPOR Conference in 2020, with an abstract and a consequent poster presentation. The abstract was then published in the Special Issue of Value in Health (DOI:https://doi.org/10.1016/j.jval.2020.08.1064). This is the rational because you have found an overlapping between the present paper and the abstract previously published. According to your kind request, we have carefully revised the text and rephrased the duplicated sentences, both in the abstract section and in the methods section of the manuscript. In this view, we confirm the originality of the present paper, since in the previous publication results have been presented partially, without any explanation of the results achieved. As mentioned above, only one single abstract have been published.

In the followings you could also find detailed reply to your questions, related to the financial disclosures.

1. Please state whether any of the research costs or authors' salaries were funded, in whole or in part, by a tobacco company (our policy on tobacco funding is at http://journals.plos.org/plosone/s/disclosure-of-funding-sources)

We confirm that the research cost or authors’ salaries are not funded by a tobacco company.

2. Please state whether the donor has any competing interests in relation to this work (see http://journals.plos.org/plosone/s/competing-interests).

The funder of the present research activity had no role in study design, data collection and analysis, decision to publish, or preparation of the manuscript. This was a decision totally dependent from the authors.

3. Please state whether the identity of the donor might be considered relevant to editors or reviewers’ assessment of the validity of the work.

The authors are not aware of any competing interests.

Replies to Reviewer #1

We would thank the Reviewer # 1 for the comments. 

1. Throughout the article you mention PVCs standard procedure. Honestly, it’s not so clear what you mean by standard. Who defined this standard? Are there any clinical guidelines or hospital guidelines? Or is it just a hypothesis you want to test? In case it’s a gold standard (the word standard implicitly refers to a gold standard) you should add a reference. If it is a standard, why is it applied only to 18% of patients? Who decides whether the standard has to be applied or not?

Thank you very much for your comment. We realized that we did not explain well the rationale behind our evaluation, as well as the word “standard” is not the correct adjective to use in this case.

The rationale behind our evaluation was a sort of a performance monitoring activities, with regard to the real-life implementation of a specific hospital PVCs management procedure, developed and proposed by the Risk Management and Quality Office of the hospitals involved with the collaboration of Nurses and Healthcare Professionals Directorate. After the approval to implement the hospital PVCs management procedure from the Healthcare Directorate, and the related practical introduction, it was necessary to verify and evaluate the effective adoption of this procedure in terms of effectiveness, safety and efficiency profiles.

Moving on from this premise, we have accordingly modified the introduction section, to better explain the innovative PVC management process, that we have now defined as “PVCs standardized optimal procedure”. In this view, and according to literature evidence available on the topic, the simultaneous and standardized use of three specific medical devices could be able to optimize the entire management process of patients requiring a PVC. Based on the above the PVCs standardized optimal procedure consisted in the use of three different disposable medical devices, such as ChloraPrep (disposable device used for skin antisepsis), Nexiva (innovative PVC, to be inserted), and Posiflush (disposable device to be used for staying “in situ” and for washing up the patients after the catheter removal) use. On one hand, the use of ChloraPrep, implemented for skin preparation, would rapidly kill micro-organism, thus also maintaining the antimicrobial activity for at least 48 hours (Florman et al., 2007; Crosby et al., 2001), and leading to a decrease in PCV associated infections’ occurrence rate (Manoury et al., 2008; McCann et al., 2016). On the other hand, the PVC Nexiva, in the comparison with traditional cannula, incorporates a stabilization platform, which could significantly reduce the risks of developing complications, such as phlebitis, infiltration/extravasation and dislodgement (Bausone-Gazda et al., 2010; Van Zundert, 2005). Beside an increased safety profile for patients, the use of the PVC Nexiva would simultaneously offer protection to the healthcare professionals, in terms of reducing the risk of exposure associated with needlestick injury (Lee et al., 2005). Furthermore, using Posiflush at the same time, would reduce catheter-related bloodstream infections, thus improving standards of practice for catheter maintenance and management (Can & Yan, 2012).

The use of all the above mentioned three devices, should also be integrated with the following targets activities, in order to guarantee the implementation of the clinical procedure, as requested by the most diffuse Italian and European Guidelines (O’Grady et al., 2011; RCN, 2010; INS, 2016; Scales, 2008): i) number of insertion attempts lower than four; ii) PVC replacement in 96 hours; and iii) number of washes higher or equal to the number of PVC insertion.

The integration of the above targets’ activities with the use of all the three disposable medical devices represents the so-called “PVCs standardized optimal procedure”.

In the present research activity, we would like to define the incremental advantages in the clinical practice related to the implementation of the so-called PVCs standardized optimal procedure, based on the real-life practice of two hospitals (in Liguria Region) involved in the analysis. In this view, according to the study design and the inclusion/exclusion criteria for patients being enrolled in the observational prospective study, in 2019 only 18% of patients were treated coherently with the PVCs standardized optimal procedure.

It should be noted here that the results presented in the paper show a real-life picture concerning the clinical practice of the two hospitals involved, where despite the standardized optimal procedure is known and it represents an internal hospital protocol, its use is not yet standardized. According to the above, we would like to highlight the fact that our study does not present an interventional nature. We just observed the current clinical practice with the comparison of the two different scenarios (implementation of the PVC standardized optimal procedure versus not implementation of the PVC standardized optimal procedure), in the same organizational context.

2. You define as intervention the “implementation of an innovative PVCs standard procedure” and as comparator the “standard procedure is not utilized for all the patients”. What do you mean by innovative: using the innovative BD Nexive PVC or using the so-called standard procedure (i.e. the three BD devices) for all patients? Also, in line 123 what do you mean by innovative scenario? It seems to me you are simply simulating different penetration rate of PVC.

Thank you for remarking this point. We would like to confirm that the innovative procedure is represented by the so-called “PVC standardized optimal procedure”, consisting of the use of all the three disposable medical devices and the implementation of targets’ activities, along the three phases of the process: skin antisepsis, insertion and washing up.

In line 123 you could find the terms “innovative scenario”, because we simulated the hospital costs related to a different use of the so-called “PVC standardized optimal procedure”. According to the budget impact analysis methodology, a baseline scenario (the current standard of care, that in this case consists of the use of PCV standardized optimal procedure only for 18% of patients, representing the percentage of penetration in the real life setting) was compared to three different innovative scenarios (differing from the baseline because of a higher use of the PVC standardized optimal procedure, so diverging from the percentage of penetration in real life).

We have amended the revised version so that it could be clearer and easier to understand. We hope to have revised the Methods section properly.

3. It’s not very clear what you mean by “being in procedure” and “not being in procedure” groups. Please explain more clearly. Also explain how the patients are assigned to the two groups: randomly? based on clinical considerations?

Thank you for the comment. As previously mentioned in the reply at your first comment, our study does not present an interventional nature. Since it was designed as an observational study, it only observed what happened in the hospitals involved regarding the use of specific medical devices as well as to conduct specific targets’ activities.

According to the above and to the detailed explanation of the PVC standardized optimal procedure, the term “being in procedure” means that all the three disposable medical devices are correctly implemented and integrated with the targets activities recommended by guidelines available on the topic, for the three phases of the PVC management process: skin antisepsis, insertion and washing-up. On the contrary, the term “not being in procedure” refers to the use of only one or two disposable medical devices. In this case other medical devices are utilized, and not all the recommended targets’ activities are performed.

Nurses choose the devices for the PVCs management process, and not always act by selecting the three devices indicated in the standardized optimal procedure, despite the patients’ characteristics being superimposable, as demonstrated in the results section, analyzing the sample, for which there are no statistically significant differences in the two populations.

4. When you explain the goal of the study in the introduction, you say “based on these considerations”, that seems to be linked to the choice of the appropriate medical device and presenting appropriate knowledge and skills (previous sentence). None of these elements is mentioned in the article. But then, in the regression analysis you mention PVC typology, so one can argue that nurses have to possibility to choose the type of device and the three BD devices are not used for all the patients. Please explain.

Thank you for your consideration. We have coherently revised the Introduction and the Methods sections following your consideration.

The present study does not have an interventional nature: despite the standardized optimal procedure was well-known in the hospitals involved, thus being integrated in the standard clinical practice, and presented as a hospital procedure, nurses would voluntarily choose the medical devices to be used in the skin antisepsis, insertion and washing up phases of the PVCs management process, having different type of devices available for these activities, thus following only partially, or completely diverging from the standardized implant procedure (requiring the use of the three cited devices, in the three process phases).

For instance, the Beta coefficient related to the variable “disinfection” refers to the different typology of medical devices used for skin antisepsis or skin preparation activity, that were numerically coded (in the amended table, we have modified the variable in “medical devices used for skin antisepsis”. According to the above, a lower number refers to the use of ChloraPrep, as disposable device for skin antisepsis activities. In this case, the use of ChloraPrep is significantly related to the achievement of a higher outcome for the patients.

5. In the abstract you mention ABC was performed, but this is not mentioned in the body of the article. The method used to collect the data in table 5 should be explained. You should also explain the meaning of each variable in table 5.

Thank you for remarking this point. We have accordingly revised the Methods section, with regard not only the implementation of the ABC methodology, but also with regard the methods used for data collection.

6. In table 1 you mention Vein Characteristics - visible and palpable: how is this variable measured? Is there an objective measure or it relies on the nurse perception? Does the operator experience influence the evaluation of this aspect?

Thank you for this comment. The vein characteristics and the classification of the veins was made according to the A-DIVA (Adult Difficult Intravenous Access) Scale (Van Loon et al., 2016), a specific tool validated in literature evidence, whose utilization is not influenced by the healthcare professionals experience. This scale was developed with the main aim to prospectively identify patients with a high probability of a difficult intravenous access based on easily available clinical data, which may improve clinical practice and patient’s comfort.

The A-DIVA scale can be applied across clinical settings and patient cohorts basing its assessment on five items: (i) palpability of the target vein; (ii) the patient’s history of difficult intravenous access; (iii) visibility of the target vein; (iv) patients’ access in a department emergency area, in terms of unplanned indication for surgery; (v) the diameter of the target vein is less than 2 millimeters. Each confirmed item adds one point to the scale’s score (ranging from 0 to 5), where a higher score indicates a higher risk of difficult intravenous access and risk of failed PVC insertion. According to this score, patients could be classified in three different groups: 1) patients considered at low risk to experience a failed PVC insertion (score range: 0-1); 2) patients considered at medium risk to experience a failed PVC insertion (score range: 2-3); 3) patients considered at high risk to experience a failed PVC insertion (score range: 4-5).

7. Please clarify which patients’ comorbidities are considered in the analysis.

Thank you for this consideration. Since the present research activity was designed as a prospective observational study, all the potential comorbidities developed by the patients enrolled, were included. In fact, the typology of comorbidities was not considered neither as an inclusion or an exclusion criterion.

According to the above, the distribution of patients concerning the typology of comorbidities is reported in the following table, considering only the 72.90% of the overall sample presenting a concomitant disease.

Typology of comorbidities %

Cardiovascular disease 31.70%

Pulmonary disease 21.20%

Neurological disease 11.40%

Oncological disease 9.10%

Infectious disease 6.30%

Psychiatric disease 6.30%

Renal disease 5.80%

Rheumatological disease 3.00%

Integumentary System disease 2.10%

Diabetes 1.90%

Musculoskeletal disorders 1.20%

8. Five wards were belonging to two different hospitals participated in the study. Did you find any differences (e.g. % of in procedure patients) among hospitals and medical vs surgical wards?

Thank you for the comment. We involved two hospitals referring to the same organizational structure. Before implementing our analyses to achieve the primary study’ objective, a preliminary assessment was performed to understand potential “ex-ante” differences among groups. We confirm that no differences emerged, so that both hospitals and wards were comparable. The habit of the nurses in the two hospitals of using the standardized optimal procedure, did not change significantly, as well as in the different medical and surgical departments.

9. In table 4 please make clear the dependent variable. Moreover, in the text you should explain more clearly the meaning of the coefficients: for instance, how to interpret the meaning of the coefficient of disinfection?

Thank you for your consideration. We have amended the Results section, referred to Table 4. In particular, the dependent variable was the effectiveness of the PVC standardized optimal procedure, in terms of PVCs removal reasons.

In general terms, the two regression analyses performed and reported in Table 4 and in Table 6, considered effectiveness and costs acting as dependent variables of the model. These variables could be influenced by the following independent variables: i) the implementation or not of the PVC standardized optimal procedure; ii) typology of medical devices used for skin antisepsis, implant and washing-up activities, considering not only the disposable medical devices composing the standardized optimal procedure, but also other comparable medical devices; iii) patient’s BMI; iv) vein characteristics; v) number of PVCs days in situ; vi) PCV-related activities execution time.

10. What do you mean by “simulated the healthcare expenditure sustained up to 12 months?” It seems to me you mean costs borne by the hospitals instead of healthcare expenditure. Purchasing cost?

Thank you for the comment. We have accordingly revised the sentence, since we have currently assumed the hospital point of view and we have defined all the costs directly sustained by hospitals in performing such procedure. Besides the purchasing costs of the medical devices investigated, we economically evaluated all the hospital clinical pathway costs of a patients requiring PVC management.

11. The costs that have been considered in the economic analysis are not clear. Where can I find the cost of the devices (three in the treatment group, 1 or 2 in the control)? Any costs for the management of adverse events?

Thank you for the comment. According to your kind request, we have modified the Methods section, to better define the economic analysis conducted. In addition, we confirm that we have included in the analysis also the cost for the management of adverse events. In Table 5 you could find both costs related to need of PVC repositioning due to the development of any adverse events, as well as economic evaluation to manage the adverse events occurred.

12. Budget impact analysis: it’s not clear to me how the numbers in table 7 were computed. In the economic?

Thank you for remarking this point. We have tried to better explain the rationale behind the budget impact analysis development, in the Method section.

In particular, we have compared the current standard of care in the hospitals involved with different innovative scenarios, presenting a different use of the standardized optimal procedure. The base-case scenario, that in Table 7 is labelled as Scenario A, consisting of the use of the PVC standardized optimal procedure only for 18% of patients, resulting from the analysis of the real life. The residual 82% presented an overall economic absorption as for “not being in a standardized optimal procedure), that is equal to € 21.71 (versus € 19.60 for those being in a standardized optimal procedure). The above base-case scenario (Scenario A) was compared to Scenario B, C and D, presenting a higher implementation rate of the PVC standardized optimal procedure.

- Innovative Scenario B refers to the implementation of the PVC standardized optimal procedure for 100% of patients requiring a PVC, with a cost per patient equal to € 19.60. Scenario B represents a sort of “best-case scenario”, in which the standardized optimal procedure has a total percentage of penetration in the clinical practice.

- Innovative Scenario C refers to the implementation of the PVC standardized optimal procedure for 50% of patients requiring a PVC (with a cost per patient equal to € 19.60). For the other 50%, the standardized optimal procedure is not implemented, thus presenting a cost per patient equal to € 21.71.

- Innovative Scenario D refers to the implementation of the PVC standardized optimal procedure for 35% of patients requiring a PVC (with a cost per patient equal to € 19.60). For the other 65%, the standardized optimal procedure is not implemented, thus presenting a cost per patient equal to € 21.71.

All the above innovative Scenarios were thus compared to Scenario A, considering the overall PVC implants performed on an annual basis by the hospitals involved, thus being equal to 156,624.

The same above Scenarios were considered with reference to the analysis of the organizational impact. In this regard, despite conducting the analysis with euros spent for patients, we conducted the analysis, in terms of system capacity, considering the time spent (in minutes) for performing all the PVC-related activities.

13. Last, I can guess that the BD products mentioned in the paper are purchased through regional tenders. In case another economic operator will award next tender, will the results be valid anyhow? On other words, are the results product-specific o procedure-specific?

Thank you for your comment. We confirm that the different medical devices comprising the standardized optimal procedure for PVC management are purchased through regional tenders. All the results could be replicable also for other economic operators, in terms of other medical devices Companies, but only if the technical characteristics of the disposable medical devices are superimposable.

14. Lines 74-79: the paragraph is too long and as a result not clear

Thank you for remarking this point. We have accordingly revised the paragraph.

15. Not very clear what BD Chloraprep and Posiflush are.

Chloraprep is a specific disposable medical device used for skin preparation and antisepsis, presenting the main aim to ensure an optimal skin disinfection and maintenance of sterility during the procedure. Chloraprep is composed of 2% chlorhexidine gluconate and 70% isopropyl alcohol in single-dose applicator and sterile solution.

On the other hand, Posiflush is a specific pre-filled syringe, containing 0.9% sodium chloride, presenting the main aim to eliminate any blood reflux and maintain the patency of the cannula, also reducing the risk of contamination. This specific medical device is used for washing-up the patients, after the PVC removal.

We have added detailed information with regard to both disposable medical devices (Chloraprep and Posiflush) in the amended manuscript, to better explain the devices characteristics.

16. What do you mean by “BD Nexiva innovative PVC”? The presence of a stabilization platform? What is this?

Thank you for your comment. We realized to have used the term “innovative” in an unproper manner. We have revised the manuscript accordingly. 

In particular, the specific feature of the PVC NEXIVA is the presence of a stabilization platform or feature, an extension set and a needle-less access site. According to the above, because PVC insertion is not considered a sterile procedure, the pre-assembly of these components greatly reduces the risk of accidental contamination of the device during the process, which could lead to bloodstream infections.

17. What do you mean by “reducing the risk of exposure associated with needlestick injury”?

Since the PVC NEXIVA represents a closed intravenous (IV) catheter system, its main features exposed above, besides preventing patients from complications or bloodstream infections, would limit healthcare professionals’ exposure blood, limiting the risk of needlestick injuries. In fact, this safety-engineered system, is also designed to reduce needlestick injury by using passive needle-shielding technology that does not compromise the insertion techniques.

18. Line 102 perspective: do you mean prospective?

Thank you for the comment. We are so sorry for the typo. We have amended the revised manuscript with term “prospective” instead of “perspective”.

19. Line 106: sign instead of signed (same in the abstract).

Thank you for the comment. We are so sorry for the typo. We have amended the revised manuscript with term “sign” instead of “signed”.

20. Line 120-121 What do you mean by “assuming 156,624 PVCs”? what type of assumption is this? Is it the volume of PVCs in the 5 analyzed departments?

Thank you for remarking this point. We have revised the sentence, to make it clearer. Actually, we refer to 156,624 PVC inserted on an annual basis, as the volume of PVC in the hospitals involved in the analysis.

21. You say “Real-life data revealed that, for patients being in procedure, 86.8% of the PVC removal was due to the end of the therapy, and not associated to adverse events, as in the other scenario (13.2%, p-value=0.000)” It’s not clear to me what you mean by “as in the other scenario”: 13from the table it seems that 13.2% relates to the same scenario, to removal due to adverse events.

Thank you for the comment, and we are so sorry for this typo. We have revised the entire sentence, because the other scenario refers to the absence of implementation of the investigated standardized optimal procedure for PVC management. We also have corrected the proper percentage rate, now equal to 39.4%.

22. It’s not clear who is the sponsor of the study.

The Principal Investigator of the study was one of the authors, Roberta Rapetti, who also proposed the implementation of the hospital standardized optimal procedure inside the hospital and who wanted, few years away from the standardized optimal procedure implementation, to validate the efficiency and effectiveness of the procedure proposed, asking for the support of LIUC for the methodological part related to the Danish Mini HTA implementation.

Replies to Reviewer #2

We would thank the Reviewer # 2 for the comments. 

1. There is no control group and sample size are too small without any theoretical sample size estimation. This may also be the reason why most of the statistical results are not significantly different. 

Thank you very much for your comment. Despite the small sample size, in our opinion results could be replicable and successful, thus demonstrating the feasibility of the adoption of a standardized optimal procedure for PVC management, both from an economic and an effectiveness point of view.

With regard to the sample size, we have deeply analyzed the sample of patients to enroll in the present study. In particular, based on a research conducted in 2017 by the hospital involved, considering a confidence interval of 99% and a power of the sample equal to 90%, our expectation was to enroll 380 patients within the present observational study. In particular, the following inclusion criteria were considered: i) age older than 18 years; ii) length of hospitalization ranging from 4 to 15 days; and iii) use of PVCs.

The informed consent form was collected, and patients who did not sign the form were excluded from the study.

2. How consistent is the assessment of EMR and ultrasound operation between different researchers? Performing a test of internal and external consistency of the researcher could be considered.

Thank you for the comment. In our research setting, EMR was not considered for the evaluation of the patients and of the standardized optimal procedure.

3. Author should consider and analysis that the working years and professional abilities of ER nurses may also be the key factors that affect the placement of PIVC and the judgment of PIVC failure or survival.

Thank you for your kind suggestion. It could be surely a topic of further research to consider the healthcare professionals seniority and the professional abilities, to be included in the regression models both for costs and for effectiveness.

4. The definition of subcutaneous edema in the image needs more precisely descript rather just presents the photo. Some parts in the article are not clearly described.

Thank you for your comment. We did not present photos, because we did not consider the subcutaneous edema in the insertion of the PVC or in the PVC removal.

5. There are inconsistent descriptions of PIVC failures in the text, such as line 164 and line 221, which may cause confusion for readers.

Thank you for your comment. We have accordingly modified the introduction and methods section, in order to better define the advantages related to PVC NEXIVA, as a closed IV catheter system capable to reduce the risk of accidental contamination of the device during the process, which could lead to bloodstream infections.

6. The primary outcome is not clear enough in line 219. What is the definition of “increased risk of PIVC failure” and how to measure it?

Thank you for your comment. We have revised the amended manuscript, detailing the primary objective of the analysis. In particular, the present study aims at analyzing the process of management of PVCs in the clinical practice, considering the hospital point of view, in relation with outcome measures achieved, in the standardized implementation of an optimal procedure for PVCs implant. In addition, both the organizational and economic incremental benefits were defined, to understand optimization area for the hospitals taking in charge patients requiring a PVC.

7. “Day idle”, a significant variable between groups in the outcome measurement, but there is no any discussion about that.

Thank you for your kind suggestion. It could be surely a topic of further research to consider also the “day-idle” variable to be included in the regression models both for costs and for effectiveness.

8. If the ultrasonographic shows subcutaneous edema, but the clinical standard assessment process by ER nurse is not abnormal, is it necessary to remove the PIVC preventively? Are there any other suggestions for the above condition? I don’t see any discussion for that.

Thank you for your comment. The assessment process by the ER nurse is not applicable in the present investigated setting, because the insertion of the PVC and the management process start in the medical or surgical departments and not in the ER.

---

## [Editor Report · Decision Letter 1]

5 Oct 2021

PONE-D-21-00993R1Organizational, efficiency and effectiveness impacts of a standardized optimal procedure, for the PVCs managementPLOS ONE

Dear Dr. Ferrario,

Thank you for submitting your manuscript to PLOS ONE. After careful consideration, we feel that it has merit but does not fully meet PLOS ONE’s publication criteria as it currently stands. Therefore, we invite you to submit a revised version of the manuscript that addresses the points raised during the review process.

Please revise the whole manuscript carefully, including the Title.

We look forward to receiving your revised manuscript.

Kind regards,

Academic Editor

PLOS ONE

Journal Requirements:

Additional Editor Comments (if provided):

What is "PVC"? Don't use acronyms or abbreviation upfront at the Title! Readers have no idea of your Title!
---

## [Author Response · Author response to Decision Letter 1]

11 Oct 2021

Dear Editor,

Thank you for your kind comment and suggestion with regard to the modification of the title of our manuscript. According to your kind request, we have revised the title from “Organizational, efficiency and effectiveness impacts of a standardized optimal procedure, for the PVCs management” to “The implementation of a standardized optimal procedure for peripheral venous catheters’ management: results from a multi-dimensional assessment”.

---

## [Decision Letter · Decision Letter 2]

24 Nov 2021

PONE-D-21-00993R2The implementation of a standardized optimal procedure for peripheral venous catheters’ management: results from a multi-dimensional assessmentPLOS ONE

Dear Dr. Ferrario,

Thank you for submitting your manuscript to PLOS ONE. After careful consideration, we feel that it has merit but does not fully meet PLOS ONE’s publication criteria as it currently stands. Therefore, we invite you to submit a revised version of the manuscript that addresses the points raised during the review process.

Please revise.

We look forward to receiving your revised manuscript.

Kind regards,

Academic Editor

PLOS ONE

Reviewers' comments:

Reviewer's Responses to Questions

**Comments to the Author**

1. If the authors have adequately addressed your comments raised in a previous round of review and you feel that this manuscript is now acceptable for publication, you may indicate that here to bypass the “Comments to the Author” section, enter your conflict of interest statement in the “Confidential to Editor” section, and submit your "Accept" recommendation.

Reviewer #3: (No Response)

2. Is the manuscript technically sound, and do the data support the conclusions?

Reviewer #3: Partly

3. Has the statistical analysis been performed appropriately and rigorously? 

Reviewer #3: Yes

4. Have the authors made all data underlying the findings in their manuscript fully available?

Reviewer #3: Yes

5. Is the manuscript presented in an intelligible fashion and written in standard English?

Reviewer #3: No

6. Review Comments to the Author

Reviewer #3: The introduction, methods, results and discussion sections need to be reviewed and properly written in good English. The method section do not clearly explain the scenario 2.

7. PLOS authors have the option to publish the peer review history of their article (what does this mean?). If published, this will include your full peer review and any attached files.

Reviewer #3: **Yes: **Henry Egi Aloh

---

## [Author Response · Author response to Decision Letter 2]

16 Dec 2021

Dear Dr. Henry Egi Aloh,

We thank you for the revisions made to our manuscript.

According to your kind suggestions, we confirm we have revised the entire manuscript to make it presented in an intelligible fashion and written in standard English. 

In addition, we have clarified and better explained the investigated technological scenarios, with specific detailed for Scenario 1 and Scenario 2.

Scenario 1 refers to the adoption of a “PVCs standardized optimal procedure”, that is the one suggested and implemented in the hospitals referring to Liguria Region (Italy) where the analysis was conducted. Scenario 1 refers to the simultaneous use of three specific medical devices suggested by the main guidelines available on the topic (O’Grady et al., 2011; RCN, 2010; INS, 2016; Scales, 2008): ChloraPrep (disposable device used for skin antisepsis), Nexiva (innovative PVC, to be inserted), and Posiflush (disposable device to be used for staying “in situ” and for washing up the patients after the catheter removal). The use of ChloraPrep, implemented for skin preparation, would rapidly kill micro-organism, thus also maintaining the antimicrobial activity for at least 48 hours (Florman et al., 2007; Crosby et al., 2001), and leading to a decrease in PCV associated infections’ occurrence rate (Manoury et al., 2008; McCann et al., 2016). The PVC Nexiva, in the comparison with traditional cannula, incorporates a stabilization platform, which could significantly reduce the risks of developing complications, such as phlebitis, infiltration/extravasation and dislodgement (Bausone-Gazda et al., 2010; Van Zundert, 2005). Beside an increased safety profile for patients, the use of the PVC Nexiva would simultaneously offer protection to the healthcare professionals, in terms of reducing the risk of exposure associated with needlestick injury (Lee et al., 2005). Using Posiflush at the same time, would reduce catheter-related bloodstream infections, thus improving standards of practice for catheter maintenance and management (Can & Yan, 2012).

The use of all the above mentioned three devices, should also be integrated with the following targets activities, to guarantee the implementation of the clinical procedure, as requested by the most diffuse Italian and European Guidelines (O’Grady et al., 2011; RCN, 2010; INS, 2016; Scales, 2008): i) number of insertion attempts lower than four; ii) PVC replacement in 96 hours; and iii) number of washes higher or equal to the number of PVC insertion.

On the contrary, Scenario 2 (i.e., not being in a standardized optimal procedure) refers to the use of only one or two out of the three disposable medical devices (thus using different devices for antisepsis, insertion, and washing-up phases), with target activities missing.

For being clearer and more comprehensive, we have produced a table of synthesis of the two scenarios investigated that you could find in the response letter.

With best regards,

The Authors

---

## [Decision Letter · Decision Letter 3]

18 Jan 2022

The implementation of a standardized optimal procedure for peripheral venous catheters’ management: results from a multi-dimensional assessment

PONE-D-21-00993R3

Dear Dr. Ferrario,

We’re pleased to inform you that your manuscript has been judged scientifically suitable for publication and will be formally accepted for publication once it meets all outstanding technical requirements.

Kind regards,

Academic Editor

PLOS ONE

Additional Editor Comments (optional):

Reviewers' comments:

Reviewer's Responses to Questions

**Comments to the Author**

1. If the authors have adequately addressed your comments raised in a previous round of review and you feel that this manuscript is now acceptable for publication, you may indicate that here to bypass the “Comments to the Author” section, enter your conflict of interest statement in the “Confidential to Editor” section, and submit your "Accept" recommendation.

Reviewer #4: All comments have been addressed

Reviewer #5: All comments have been addressed

2. Is the manuscript technically sound, and do the data support the conclusions?

Reviewer #4: Yes

Reviewer #5: Yes

3. Has the statistical analysis been performed appropriately and rigorously? 

Reviewer #4: Yes

Reviewer #5: Yes

4. Have the authors made all data underlying the findings in their manuscript fully available?

Reviewer #4: Yes

Reviewer #5: Yes

5. Is the manuscript presented in an intelligible fashion and written in standard English?

Reviewer #4: Yes

Reviewer #5: Yes

6. Review Comments to the Author

Reviewer #4: (No Response)

Reviewer #5: The implementation of a standardized optimal procedure for peripheral venous

catheters’ management: results from a multi-dimensional assessment

Manuscript Number: PONE-D-21-00993R3

The author has made substantial modification in this manuscript.

7. PLOS authors have the option to publish the peer review history of their article (what does this mean?). If published, this will include your full peer review and any attached files.

Reviewer #4: No

Reviewer #5: No

---

## [Editor Report · Acceptance letter]

20 Jan 2022

PONE-D-21-00993R3 

The implementation of a standardized optimal procedure for peripheral venous catheters’ management: results from a multi-dimensional assessment 

Dear Dr. Ferrario:

I'm pleased to inform you that your manuscript has been deemed suitable for publication in PLOS ONE. Congratulations! Your manuscript is now with our production department. 

Kind regards, 

on behalf of

Dr. Robert Jeenchen Chen 

Academic Editor

PLOS ONE